# Interspecies Diversity of Osmotic Gradient Deformability of Red Blood Cells in Human and Seven Vertebrate Animal Species

**DOI:** 10.3390/cells11081351

**Published:** 2022-04-15

**Authors:** Adam Varga, Adam Attila Matrai, Barbara Barath, Adam Deak, Laszlo Horvath, Norbert Nemeth

**Affiliations:** 1Department of Operative Techniques and Surgical Research, Faculty of Medicine, University of Debrecen, Moricz Zsigmond u. 22, H-4032 Debrecen, Hungary; varga.adam@med.unideb.hu (A.V.); matrai.adam@med.unideb.hu (A.A.M.); barath.barbara@med.unideb.hu (B.B.); deak.adam@med.unideb.hu (A.D.); 2Department of Pharmaceutical Surveillance and Economics, Faculty of Pharmacy, University of Debrecen, Nagyerdei krt. 98, H-4032 Debrecen, Hungary; horvath.laszlo@pharm.unideb.hu

**Keywords:** hemorheology, red blood cell deformability, osmotic gradient deformability, osmoscan, interspecies diversity

## Abstract

Plasma and blood osmolality values show interspecies differences and are strictly regulated. The effect of these factors also has an influence on microrheological parameters, such as red blood cell (RBC) deformability and aggregation. However, little is known about the interspecies differences in RBC deformability at various blood osmolality levels (osmotic gradient RBC deformability). Our aim was to conduct a descriptive–comparative study on RBC osmotic gradient deformability in several vertebrate species and human blood. Blood samples were taken from healthy volunteers, dogs, cats, pigs, sheep, rabbits, rats, and mice, to measure hematological parameters, as well as conventional and osmotic gradient RBC deformability. Analyzing the elongation index (EI)–osmolality curves, we found the highest maximal EI values (EI max) in human, dog, and rabbit samples. The lowest EI max values were seen in sheep and cat samples, in addition to a characteristic leftward shift of the elongation index–osmolality curves. We found significant differences in the hyperosmolar region. A correlation of mean corpuscular volume and mean corpuscular hemoglobin concentration with osmoscan parameters was found. Osmotic gradient deformability provides further information for better exploration of microrheological diversity between species and may help to better understand the alterations caused by osmolality changes in various disorders.

## 1. Introduction

Hemodynamics and blood flow characteristics are partly determined by hemorheological parameters, such as blood and plasma viscosity, red blood cell (RBC) deformability, and aggregation. These latter, so-called microrheological factors play a pivotal role in forming the blood’s non-Newtonian feature and in determining microcirculation and tissue perfusion [1,2]. Hemorheological parameters show alterations in numerous pathophysiological processes and disorders [3,4,5,6,7,8,9] and represent interspecies differences as well [10,11,12,13,14,15,16]. Studying these parameters thus provides important information on blood flow properties in clinical investigations and biomedical studies, as well as in zoological and veterinary sciences. Despite many experimental and clinical studies, the mechanisms of hemorheological alterations, the detailed background of microrheological features, and the very colorful and bewildering interspecies differences have not been completely elucidated yet.

One of the most widely studied microrheological parameters is RBC deformability, which is the passive ability of cells to deform by shearing and compressing external forces [1,17,18]. This ability is essential to pass through narrow capillaries, but also contributes to the viscoelastic properties of the blood at higher shear rates of flow [1,2]. RBC deformability can be influenced by several factors: cell morphology, surface-to-volume ratio, cell membrane viscosity and elasticity, and intracellular viscosity [17,18,19,20]. Important modulating factors include oxygenation level, blood pH lactate concentration, and osmolality, in addition to many other plasmatic and metabolic factors [3,9,17,20,21,22,23]. Therefore, it is not surprising that RBC deformability can be impaired in numerous hematological disorders affecting the morphology, size, membrane composition, and hemoglobin content of cells (e.g., hereditary spherocytosis, various thalassemias, hemoglobinopathies, sickle cell anemia, etc.), in inflammatory processes (sepsis, ischemia–reperfusion), and in malignancies and metabolic and cardio–cerebrovascular disorders [3,4,5,6,9,24].

Due to the main determining and modulating factors mentioned above, RBC deformability also shows a wide diversity among animal species [12,13]. However, the exact explanation or theory that would uncover the detailed background of this diversity is still unclear. It is well known that osmolality affects RBC volume as cells are swelling or shrinking, thus affecting the surface-to-volume ratio and deformability, in addition to the importance of the mechanical and elastic properties of the membrane. Blood and plasma osmolality is strictly regulated and also shows interspecies differences. It has also been widely reported that osmotic fragility differs in animal species of various vertebrate classes, ecosystems, and even positions in the food chain (herbivores, omnivores, carnivores) [25,26,27,28]. The osmotic fragility test examines the percentage of hemolysis in the function of osmolality, but it cannot determine anything about the entire process of swelling–shrinking (without rupture) and changes in deformability along an osmolality gradient. However, the question arises as to whether the change in the dynamics of RBC deformability change in the function of osmolality alteration may also show interspecies diversity.

RBC deformability can be measured by ektacytometry by investigating cell elongation in the function of applied shear stress (elongation index–shear stress curves) [29,30]. In the case of osmotic gradient ektacytometry (osmoscan), the method can provide information on the change in RBC deformability changing in the function of osmolality at constant shear stress (elongation index–osmolality curves) [31]. The method is a sensitive way of analyzing the deformability of RBCs over a wide osmolality range and studying their mechanical behavior under hypo- and hyperosmolar conditions, and in between, while also providing data on the ‘optimal’ osmolality values where RBCs have the best elongation/deformation ability. Only one point of the entire elongation index–osmolality curve corresponds to the osmotic fragility test parameter. The method is useful for supporting the clinical diagnosis of various hematological disorders (e.g., inherited RBC membrane disorders), and increasing data support alterations in osmotic gradient RBC deformability in sepsis, ischemia–reperfusion models, and metabolic disorders [21,32,33,34]. However, very few papers have dealt with the comparative aspects. Our group was the first to conduct comparative studies on RBC osmotic gradient deformability in laboratory animal species, according to laboratory standards [14,32,35].

We hypothesized that the osmotic gradient deformability of red blood cells may show wider diversity in experimental/laboratory animal species, which can be important in extrapolation of the results found in animal versus human experiments. The motivation for conducting a descriptive–comparative study was to reveal and better understand these microrheological differences and their relation to other microrheological and hematological variables. When evaluating and comparing the results, as an extrapolation, it is important to know the species-specific characteristics of these parameters. Therefore, our aim was to extend the investigations and support the osmoscan data with hematological and conventional RBC deformability measurements in humans and seven mammalian animal species (pig, dog, cat, sheep, mouse, rat, and rabbit). These animal species are commonly or relatively commonly used in various experiments as small animal models or large animal models. To our knowledge, to date, this is the most comprehensive comparative study on osmoscan parameters and directly related variables in a wide range of animal species compared to humans.

## 2. Materials and Methods

### 2.1. Volunteer Participants, Experimental Animals, and Blood Sampling

Blood samples were collected from healthy female volunteers in the morning hours after overnight fasting (*n* = 8, age: 19–40 years; Clinical Ethical Committee Approval No.: DE-RKEB 3625-2012). The samples were taken from the antecubital vein (21 G needle, BD Vacutainer^®^ tubes, 1.5 mg/mL K3-EDTA; Becton, Dickinson and Company, Franklin Lakes, NJ, USA).

We also examined blood samples taken from male beagle dogs (*n* = 6, bodyweight: 17.05 ± 1.05 kg), male cats (*n* = 7, bodyweight: 4.25 ± 0.65 kg), female Hungahib-39 pigs (*n* = 8, bodyweight: 15.2 ± 1.1 kg), female Merino sheep (*n* = 8, bodyweight: 71 ± 5.31 kg), female C57BL/b mice (*n* = 14, bodyweight: 32.1 ± 3.8 g), female Wistar rats (*n* = 10, bodyweight: 302.8 ± 12 g), and female New Zealand white rabbits (*n* = 4, bodyweight: 3063.5 ± 230.18 g). The samples were taken from animals without any other interventions (as control groups or control samplings only). None of the animals were sacrificed solely for this comparative study. All animal experiments were carried out according to the Hungarian Animal Protection Act (Law XVIII/1998) and approved by the University of Debrecen Committee of Animal Welfare (Permission Registration No.: 24/2016/UDCAW).

Blood samples were taken from the dogs and cats via cephalic vein puncture (anesthesia: 10 mg/kg ketamine + 1 mg/kg xylazine, i.m.), from pigs via medial saphenous vein puncture (anesthesia: 15 mg/kg ketamine + 1 mg/kg xylazine, i.m.), and from sheep via external jugular vein puncture. In mice, rats, and rabbits, blood sampling was also carried out under general anesthesia (mice: 60 mg/kg thiopental, i.p., sampling site: inferior caval vein; rats: 60 mg/kg thiopental, i.p., sampling site: caudal caval vein; rabbits: 60 mg/kg thiopental, i.p., sampling site: lateral ear vein). The blood was drawn into standard Vacutainer tubes (anticoagulant: sodium-EDTA, 1.5 mg/mL). We completed all the laboratory measurements within 2 h after the blood sampling.

### 2.2. Hematological Measurements

Sysmex F-800 and Sysmex K4500 microcell counter devices (TOA Medical Electronics Co., Ltd., Kobe, Japan) were used to measure RBC count (10^12^/µL), white blood cell count (WBC, 10^9^/µL), hemoglobin concentration (Hgb, g/dL), and platelet count (Plt, 10^9^/µL). Calculated values by the automates were hematocrit (Hct, %), mean corpuscular volume (MCV, fL), mean corpuscular hemoglobin (MCH, pg), and mean corpuscular hemoglobin concentration (MCHC, g/L).

### 2.3. Red Blood Cell Deformability (Conventional and Osmotic Gradient Ektacytometry)

RBC deformability was determined using a LoRRca Maxsis Osmoscan ektacytometer (RR Mechatronics International B.V., Zwaag, The Netherlands) [30,36]. In this ektacytometry method, RBCs were subjected to shear stress, and their elongation was determined by laser diffraction techniques. The so-called elongation index (EI) was determined by the function of shear stress (SS, Pa). Shear stress ranged between 0.3 and 30 Pa. For the conventional deformability test, 10 μL of whole blood was gently mixed with 2 mL of polyvinylpyrrolidone (PVP)–PBS solution (PVP: 360 kDa, Sigma-Aldrich Co., St. Louis, MO, USA; PVP-PBS solution viscosity = 28–32 mPas, osmolality = 290–310 mOsmol/kg, pH = 7.3–7.5). All measurements were carried out at 37 °C [36]. From the EI–SS curves, comparative data were used, such as EI values at 3 Pa, and by parameterization of the entire EI–SS curve, maximal elongation index (EI_max_) and shear stress at half EImax (SS_1/2_, Pa), which were calculated based on the Lineweaver–Burk equation [37].

Osmotic gradient deformability (osmoscan) measurements were carried out using 5 mL of isotonic PVP-PBS (see above) that was mixed with 250 μL of blood. In this module, the determination of EI is performed at constant shear stress (set for 30 Pa), while the osmolality of the suspension changes as the device mixes low-osmolar (0 mOsm/kg) and high-osmolar (500 mOsm/kg) PVP solutions with the whole-blood sample. The blood sample was aspirated to this PVP solution with a gradually increasing osmolality, while the elongation index was continuously registered. The result was a characteristic EI–osmolality (O) curve (Figure 1), with several notable points.

EI min represents the minimal elongation index in the low-osmolar environment. The associated osmolality value, O min (osmolality at EI min), roughly corresponds to an osmolality value where 50% of the RBSs hemolyze in the osmotic fragility test. EI max here means the maximal elongation index in the function of osmolality (note: it is not the same as EI_max_ that was calculated by the Lineweaver–Burk equation, see above). Osmolality at EI max (O (EI max) is the value where RBCs deform optimally. EI hyper (half of the maximal elongation index in the high-osmolar environment) and O hyper show the point in the hyperosmolar region where the RBCs are half of their maximal elongation. Another parameter is the Area, which is calculated from the area under the individual EI–O curves [30,32].

Besides the standard comparative parameters of the osmoscan curves, further parameters were calculated, such as ΔEI (absolute difference of maximal and minimal EI values), ΔO (absolute difference of osmolality values at maximal and minimal EI), and ratio values: EI max/EI min (rEI), O (EI max)/O min (rO), ΔEI/ΔO, and rEI/rO [32].

### 2.4. Statistical Analysis

SigmaStat Software 3.1.1.0 (Systat Software Inc., San Jose, CA, USA) was used to carry out the statistical analyses. Data are generally presented as means ± S.D. or median, 25% and 75% percentiles, and maximum and minimum values. To analyze existing differences between the species, the *t*-test or the Mann–Whitney rank-sum test was used based on the results of the normality test. One-way ANOVA tests (Bonferroni) were also used. After testing the normality of the data distribution by the Kolmogorov–Smirnov test, the correlation of the data was determined using Pearson/Spearman correlation on certain hematological and osmoscan parameters. A value of *p* < 0.05 was considered statistically significant.

## 3. Results

### 3.1. Hematological Parameters

The general quantitative and qualitative hematological parameters are summarized in Table 1.

White blood cell counts were significantly lower in rodents compared to humans (*p* < 0.001 vs. mouse, *p* < 0.001 vs. rat, *p* = 0.035 vs. rabbit). Red blood cell count was the highest in cats, followed by sheep, mice, and dogs. In dogs and cats, we found the highest hemoglobin concentration and hematocrit values. These two variables were significantly greater compared to other species, including the human blood samples as well (all: *p* < 0.001). Mean corpuscular volume and mean corpuscular hemoglobin values were significantly higher in humans. Platelet count showed higher values in almost every mammalian blood sample, except cats, which were significantly lower (all: *p* < 0.001 vs. human).

The detailed statistical differences (*p*-values) of hematological parameters between each species are summarized in the Appendix A.

### 3.2. Red Blood Cell Deformability

The elongation index (EI) in the function of shear stress (SS) plots of the different mammalian species are presented in Figure 2.

Comparing the different mammalian species, it is visible that sheep RBCs were deformed the least (all: *p* < 0.001 vs. sheep). In the low-shear section of the curve, human samples were significantly less deformed compared to the other groups, except for sheep samples (*p* < 0.05 vs. dog, cat, pig, mouse, rat, and rabbit). The comparative calculated parameters of the individual EI–SS curves are shown in Figure 3.

EI at 3 Pa results expressed significant differences between humans and the other investigated species (human vs. dog: *p* < 0.001, human vs. cat: *p* = 0.003, human vs. pig: *p* < 0.001, human vs. sheep: *p* < 0.001, human vs. mouse: *p* < 0.001, human vs. rat: *p* < 0.001, human vs. rabbit: *p* < 0.001) (Figure 3A). Rodents had the highest EI at 3 Pa levels (mouse, rat, and rabbit vs. human: *p* < 0.001). By parameterization of the EI–SS curves, we found marked differences in EI_max_ values (Figure 3B). The highest EI_max_ values were observed in rats, followed by dog and human samples. It could be seen that the ovine RBCs were deformed the least (sheep vs. all other species: *p* < 0.001). These results are strongly correlated with EI at 3 Pa results as well. The SS_1/2_ values significantly differed in the mammalian species (Figure 3C). Human SS_1/2_ results were the highest (*p* < 0.001 vs. all), as the slope of the human EI–SS curves notably altered from the other species (Figure 2). These findings were reflected well in EI_max_/SS_1/2_ values (Figure 3D).

### 3.3. Osmotic Gradient Deformability

The osmotic gradient ektacytometry representative curves of the different mammal species are shown in Figure 4.

The shape of the EI–O (osmoscan) curves showed the classic bell shape in all the investigated species. If we compare the rat curve to the human, dog, and rabbit curves, a major difference is visible at the low-osmolar region of the curves. The tendency of the pig and mouse plots was lower, and a characteristic leftward shift of the curves could be observed. The osmoscan parameters showed a great analogy between these two groups, which cannot be stated for the results of the cats and sheep. Osmoscan curves of cats and sheep were shifted to the right on the scale. The conventional comparative parameters of the osmoscan curves are presented in Figure 5 and Figure 6.

The EI min and EI max values showed great interspecies differences between the study groups (Figure 5A,B). Significant differences were found compared to human blood in dogs (*p* < 0.001), pigs (*p* < 0.001), rats (*p* = 0.004), and rabbits (*p* < 0.001). The rabbits have the highest EI min values, being significantly different from the other species, except for rats. The lowest EI min values were detectable in sheep (*p* < 0.001 vs. dog, *p* = 0.014 vs. cat, *p* < 0.001 vs. pig, rat, and rabbit). In the EI max results, these alterations were much more moderate; however, they showed interspecies diversity. In comparison to the human samples, the dog and rodent samples had higher values (dog: *p* = 0.003, mouse: *p* = 0.043), while sheep had the lowest (*p* < 0.001 vs. human, *p* = 0.047 vs. rat, *p* = 0.012 vs. rabbit). EI hyper values were relatively close to each other in the investigated mammalian species (Figure 5C), except for sheep (*p* < 0.001 vs. all). Dogs had the highest EI hyper values.

Osmolality parameters (O min, O (EI max), O hyper) also showed colorful diversity. We found the same pattern as in the elongation result (Figure 6A–C).

Significantly higher O min values were found in the sheep (*p* < 0.001 vs. all), while the dogs had the lowest values (*p* < 0.001 vs. all). Related to the O (EI max) parameter that reflects osmolality at the highest elongation index, three groups of the investigated mammalian species can be formed, arbitrarily: human and dog (human: 303.05 ± 10.92 mOsm/kg, dog: 272 ± 6.72 mOsm/kg), rodents with higher values (mouse: 327 ± 14.57 mOsm/kg; rat: 315.3 ± 18.45 mOsm/kg; rabbit: 319.75 ± 10.5 mOsm/kg), and pig, cat, and sheep with the highest values (pig: 360.88 ± 19.28 mOsm/kg, cat: 370.14 ± 19.94 mOsm/kg, sheep: 396.88 ± 15.23 mOsm/kg). The O hyper values of cats, pigs, sheep, and mice were the highest, while human and canine blood samples showed the lowest.

The Area parameter is derived from the area under the EI–O curves. Ascending order of Area parameters was found according to the following: sheep < cat < pig < mouse < human < rat < rabbit < dog.

Further comparative parameters (ΔEI, ΔO, EI max/EI min, O (EI max)/O min, and their ratios) are summarized in Table 2. These additional parameters focus on the left region of the EI–O curves, reflecting the magnitude of changes in EI in the function of osmolality at the hypo-osmolar direction.

The absolute difference in EI min and EI max values (as ΔEI) was comparable to each other; however, pig, sheep, and mouse values differed significantly from human (*p* = 0.027, *p* < 0.001, and *p* = 0.01, respectively). The difference between O min and O (EI max) (as ΔO) was markedly higher in cats, pigs, and sheep (*p* < 0.001) and moderately higher in mice, rats, and rabbits compared to humans (*p* = 0.01, *p* = 0.011 and *p* < 0.001, respectively). Their ratio (ΔEI/ΔO) was almost identical in all the investigated species. The ratio of EI max and EI min reflects the diversity that was seen in conventional red blood cell deformability measurements (Figure 2 and Figure 3) (dog: *p* < 0.001, pig: *p* < 0.001, rat: *p* = 0.008, rabbit: *p* = 0.004 vs. human). The ratio of O (EI max) and O min was very close to each other in the species; however, compared to humans, the values of dogs, pigs, rats, and rabbits differed significantly (*p* < 0.001, *p* < 0.001, *p* = 0.002 and *p* = 0.004, respectively).

Detailed statistical differences (*p*-values) of red blood cell conventional and osmotic gradient deformability parameters between each species are summarized in the Appendix A.

### 3.4. Correlation of Parameters

Since certain osmoscan parameters are related to cell volume (dominantly at hypo-osmolar region), density, and intracellular viscosity (mostly at hyperosmolar region), correlation analysis of mean corpuscular volume (MCV) and mean corpuscular hemoglobin concentration (MCHC) were also carried out on pooled data. A significant correlation was found between the following variables: MCV–EI min (coefficient: 0.414, R^2^ = 0.1714, *p* = 0.0013), MCV–O min (coefficient: −0.696, R^2^ = 0.4844, *p* < 0.001), MCV–∆O (coefficient: −0.571, R^2^ = 0.326, *p* < 0.001), MCV–Area (coefficient: 0.562, R^2^ = 0.3158, *p* < 0.001), MCHC–EI hyper (coefficient: 0.523, R^2^ = 0.2735, *p* < 0.001), and MCHC–O hyper (coefficient: −0.486, R^2^ = 0.2361, *p* = 0.0014).

## 4. Discussion

The osmoregulatory processes of living organisms are complex and have acquired their real function during evolutionary development [38]. Osmoregulation is the key process in maintaining electrolyte and water balance (osmotic balance) across cell membranes. Mammalian systems have evolved to control not only the overall osmotic state across cell membranes (including the red blood cell membrane), but also specific concentrations of important electrolytes in the three major fluid compartments: blood plasma, interstitial fluid, and intracellular fluid. The body’s water balance can shift due to several pathophysiological processes. Decreased plasma osmolality is observed in hyponatremia, hyperhydration, and SIADH. In contrast, osmolality is increased in chronic renal failure, ketoacidosis, dehydration, hypernatremia, and the presence of other exogenous substances. Differences in the shape of red blood cell osmoscan (elongation index–osmolality) curves can also be observed in various diseases and hematological diseases (e.g., hereditary spherocytosis, β-thalassemia, essential thrombocythemia, polycythemia vera, hereditary stomatocytosis, myeloproliferative hematological malignancies) [21,31,32,39,40,41].

Interspecies hemorheological differences have been widely investigated, showing colorful diversity, depending on the variables and methodology, and also without an exact or uniform explanation [7,9,12]. Interspecies differences in osmotic gradient deformability have been poorly investigated, yet little data are available in the literature [14,32,35]. In this descriptive–comparative study, the aim was to extend our previous research work by widening the investigated species with a detailed analysis of the osmoscan measurements.

Similar to the literature data and to our previous publications, we observed that red blood cells of different mammalian species deform differently under shearing forces, and the aggregation and aggregability of erythrocytes also vary between species [11,12,14,15,16]. The highest EI values were observable in mouse and canine blood, typically above shear stress of 1 Pa. In contrast, some lower values were shown in rats and lower in porcine and feline blood, while the lowest EI data were detected in sheep blood, mostly above shear stress of 3–5 Pa. In humans, sheep, and cats, the morphology of EI–SS curves differed compared to the other species (Figure 2). Interspecies diversity is observed in the elongation index values obtained during the osmotic gradient ektacytometric measurement as well.

Osmolality of the plasma and the blood [42,43] is known to show interspecies differences. Literature data are available mostly on plasma osmolality [44,45,46,47], while in our study, according to the methodology, whole-blood samples were used. Here, the O (EI max) value can be considered the normal osmolality of the blood. The normal range of human plasma osmolality is about 275–295 mOsm/kg; in our study, the O (EI max) values were 303.5 ± 10.1 mOsm/kg. Plasma osmolality normal ranges [44,45,46,47] and O (EI max) values partly differed in dog (280–300 vs. 272 ± 6.7 mOsm/kg), cat (290–340 vs. 360 ± 19.9 mOsm/kg), pig (310–340 vs. 358.2 ± 19.7 mOsm/kg), sheep (310–340 vs. 396.8 ± 15.2 mOsm/kg), mouse (310–330 vs. 324 ± 17.5 mOsm/kg), rat (300–310 vs. 315.3 ± 18.4 mOsm/kg), and rabbit (310–330 vs. 319.8 ± 10.5 mOsm/kg).

In the osmoscan curves, the point set by EI min and O min can be partly related to classical osmotic fragility tests. However, wide interspecies diversity of osmotic fragility data has been reported [25,26,27,28,48]. Here, the cells start to rupture, but the magnitude and number of fragmented cells cannot be correctly determined with this ektacytometry device. Therefore, a clear comparison with the osmotic fragility test is not possible. However, the curve part appearing leftward from the maximal EI point reflects the elastic features of the cells, as they swell with decreasing osmolality. Since the surface-to-volume ratio and shape of the cells change, their deformability decreases. Concerning interspecies differences, the cell volume, cell shape, membrane viscosity, and elastic features, as well as the water permeability of the cell membrane, can be listed as components of the variety [3,12,17,31,49,50,51]. At O (EI max), red blood cells can deform maximally at a given shear stress, representing the optimal osmolar condition for the cells. With increasing osmolality, the cell volume decreases with an increase in intracellular viscosity and density, as well as with changing cell surface-to-volume ratio. The result is a decreasing elongation and impaired deformability again. As MCHC and MCV show a wide diversity in animals, this parameter also serves as an important explanation of the osmoscan differences. It is also important to mention that osmoscan data, and even the shape of elongation index–osmolality curves, are shear-stress dependent [14,32,52]. As the applied shear stress is lowered, the osmoscan curves move leftward with depression in the hyperosmolar region. The magnitude of this phenomenon also shows interspecies differences [32]. Besides the investigated parameters above, further analysis would be important to clarify these interspecies differences. Unfortunately, there are very limited literature data on osmotic gradient ektacytometry and the explanation of these results.

The main limitations of this study include the relatively low case numbers and certain methodological conditions. Since the shape of the osmoscan curves, and thus the values, are shear-stress dependent [32], it would be interesting to investigate these variables in various shear stress settings. Due to the limitation of the blood sampling volumes, we could not extend the investigations in this matter. In human studies, the set shear stress is generally 30 Pa (or 20 Pa); therefore, for the comparison, we decided to use one setting (30 Pa). Another limitation is the number of species. We selected mammalian species used in surgical research models. A wider investigation would give more valuable information on the characteristics and relationship (even taxonomical) between interspecies diversity. A further limitation is the potential effect of the gender and age difference of the animal species that were investigated [7,53].

## 5. Conclusions

To our knowledge, this study presents the widest range of mammalian species with the most detailed osmotic gradient deformability investigations of red blood cells. We could state that the osmotic gradient deformability of red blood cells shows wide interspecies diversity. The shape of the elongation index–osmolality curves shows a colorful morphology in the investigated mammalian species compared to humans. At the hypo-osmolar region, the magnitude and dynamics of the elongation index decreasing with lowering osmolality are well balanced and comparable in the species. A correlation of mean corpuscular volume and mean corpuscular hemoglobin concentration with osmoscan parameters was found. Osmotic gradient deformability provides further information for better exploration of microrheological diversity between species and may help to better understand the alterations caused by osmolality changes in various disorders. The results may be useful for further comparative studies in biorheology. The findings might help the extrapolation and comparability of data with humans. Furthermore, studying erythrocytes from various species with different red blood cell volume, shape, and biomechanical features may also provide information on the microrheological features of abnormal cells in human hematological disorders.

## Figures and Tables

**Figure 1 cells-11-01351-f001:**
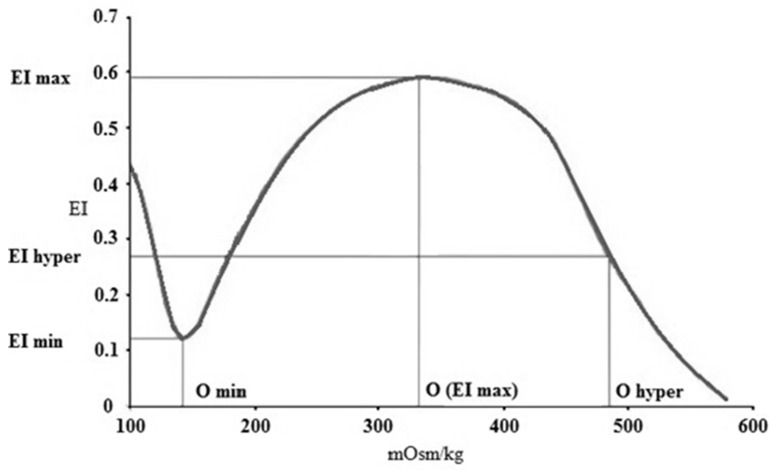
Representative elongation index (EI)–osmolality (O) (mOsm/kg) curve with the notable points. EI min (minimal elongation index), O min (osmolality at EI min), EI max (maximal elongation index), O max (osmolality at EI max), O hyper (osmolality in the hypertonic region at 50% of EI max), and EI hyper (half EI max at hyperosmolar region).

**Figure 2 cells-11-01351-f002:**
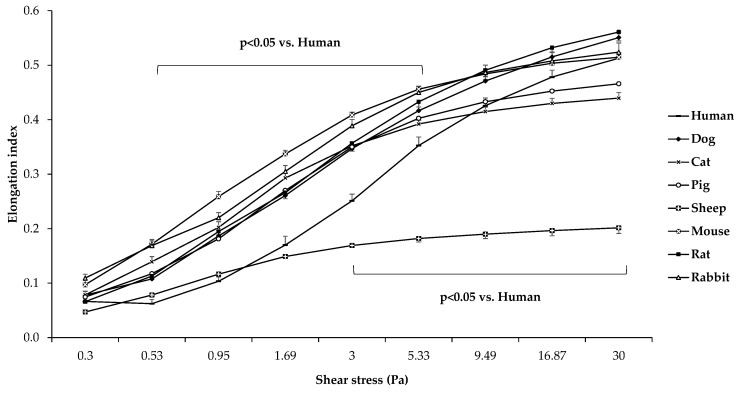
Elongation index (EI) in the function of shear stress (SS [Pa]) in human (*n* = 8), dog (*n* = 6), cat (*n* = 7), pig (*n* = 8), sheep (*n* = 8), mouse (*n* = 14), rat (*n* = 10), and rabbit (*n* = 4) blood samples. Means ± S.D.

**Figure 3 cells-11-01351-f003:**
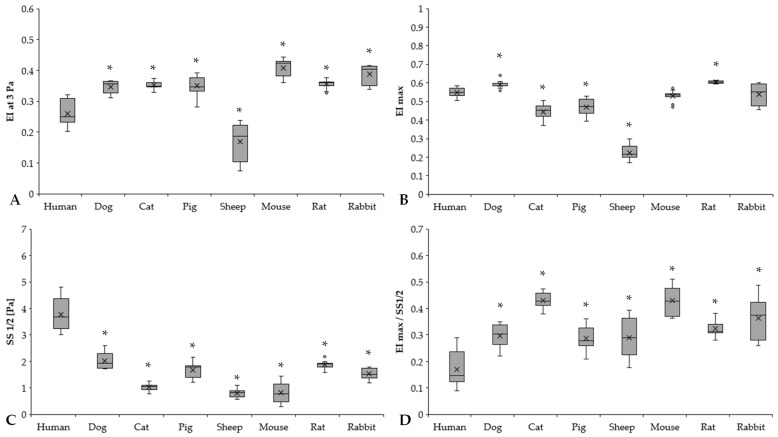
Elongation index at 3 Pa (EI at 3 Pa) (**A**), calculated maximal elongation index (EI_max_) (**B**), shear stress at half EI_max_ (SS_1/2_, Pa) (**C**), and EI_max_/SS_1/2_ ratio (**D**) of human (*n* = 8), dog (*n* = 6), cat (*n* = 7), pig (*n* = 8), sheep (*n* = 8), mouse (*n* = 14), rat (*n* = 10), and rabbit (*n* = 4) blood samples. Means ± S.D., * *p* < 0.05 vs. human.

**Figure 4 cells-11-01351-f004:**
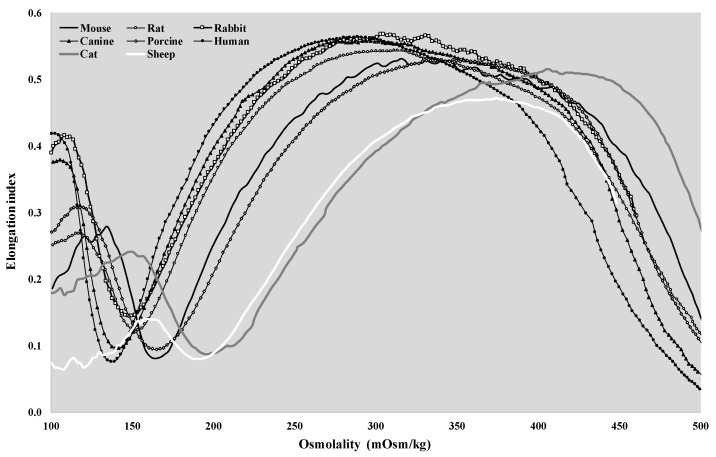
Representative osmoscan plots of human, canine, feline, porcine, ovine, murine, rat, and rabbit blood samples.

**Figure 5 cells-11-01351-f005:**
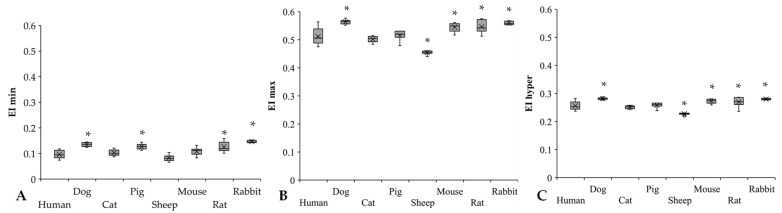
EI min (**A**), EI max (**B**), and EI hyper (**C**) results of human (*n* = 8), dog (*n* = 6), cat (*n* = 7), pig (*n* = 8), sheep (*n* = 8), mouse (*n* = 14), rat (*n* = 10), and rabbit (*n* = 4) blood samples. Means ± S.D., * *p* < 0.05 vs. human.

**Figure 6 cells-11-01351-f006:**
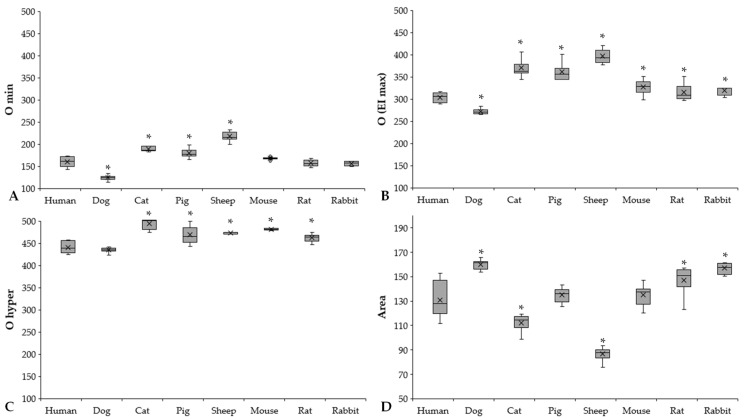
O min (**A**), O (EI max) (**B**), O hyper (**C**) (mOsm/kg), and Area (**D**) results of human (*n* = 8), dog (*n* = 6), cat (*n* = 7), pig (*n* = 8), sheep (*n* = 8), mouse (*n* = 14), rat (*n* = 10), and rabbit (*n* = 4) blood samples. Means ± S.D., * *p* < 0.05 vs. human.

**Table 1 cells-11-01351-t001:** Hematological parameters in human (*n* = 8), dog (*n* = 6), cat (*n* = 7), pig (*n* = 8), sheep (*n* = 8), mouse (*n* = 14), rat (*n* = 10), and rabbit (*n* = 4) blood samples.

Variable	Human	Dog	Cat	Pig	Sheep	Mouse	Rat	Rabbit
WBC (G/L)	6.13 ± 0.85	11.76 ± 2.33 *	10.1 ± 4.34 *	12.93 ± 3.05 *	145.93 ± 37.13 *	4.85 ± 0.99 *	2.57 ± 0.5*	5.41 ± 0.28 *
RBC (T/L)	4.57 ± 0.21	8.16 ± 0.51 *	11.52 ± 0.89 *	6 ± 0.56 *	10.41 ± 0.98 *	8.3 ± 0.41 *	5.95 ± 0.55 *	6.02 ± 0.27 *
Hgb (g/L)	127.69 ± 10.16	185.83 ± 7.73 *	153.54 ± 14.22 *	97.44 ± 8.61 *	115.75 ± 12.23 *	131.6 ± 6.55	108.8 ± 3.61 *	127.5 ± 4.47
Hct (%)	42.65 ± 1.62	55.65 ± 2.75 *	56.3 ± 5.89 *	35.23 ± 3.13 *	43.48 ± 4.19	44.6 ± 2.36 *	36.48 ± 4.23 *	40.31 ± 1.48 *
MCV (fL)	93.4 ± 1.65	68.2 ± 1.26 *	48.85 ± 2.9 *	59.56 ± 3.11 *	41.77 ± 0.9 *	53.72 ± 0.44 *	61.17 ± 1.96 *	66.84 ± 1.51 *
MCH (pg)	28.04 ± 2.12	22.78 ± 0.61 *	13.33 ± 0.6 *	16.36 ± 1.75 *	11.11 ± 0.41 *	15.87 ± 0.24 *	18.41 ± 1.68 *	21.15 ± 0.72 *
MCHC (g/L)	309.5 ± 31.2	334.25 ± 4.71	273.31 ± 10.06 *	238.56 ± 45.9 *	266.31 ± 13.59 *	295.05 ± 4.45	301.85 ± 31.29	316.25 ± 3.2
Plt (G/L)	260.38 ± 28.66	411.67 ± 100.43 *	396.75 ± 38.51	641.88 ± 135 *	413.5 ± 0.71 *	880.05 ± 114.38 *	686.45 ± 118.2 *	481.88 ± 13.56 *

WBC: white blood cell count; Hct: hematocrit; RBC: red blood cell count; Hgb: hemoglobin; MCV: mean corpuscular volume; MCH: mean corpuscular hemoglobin; MCHC: mean corpuscular hemoglobin concentration; Plt: platelet count. Means ± S.D.; * *p* < 0.05 vs. human.

**Table 2 cells-11-01351-t002:** Further comparative parameters of elongation index (EI)–osmolality (O) curves calculated from the conventional osmoscan data in human (*n* = 8), dog (*n* = 6), cat (*n* = 7), pig (*n* = 8), sheep (*n* = 8), mouse (*n* = 14), rat (*n* = 10), and rabbit (*n* = 4) blood samples.

Variable	Human	Dog	Cat	Pig	Sheep	Mouse	Rat	Rabbit
ΔEI	0.42 ± 0.02	0.43 ± 0.01	0.41 ± 0.02	0.39 ± 0.02 *	0.37 ± 0.01 *	0.47 ± 0.06 *	0.41 ± 0.04	0.41 ± 0.01
ΔO	143.5 ± 4.87	147.33 ± 10.29	180.2 ± 20.59 *	181 ± 12.01 *	179.25 ± 12.24 *	158.6 ± 13.78 *	157.8 ± 13.38 *	163 ± 6.48 *
ΔEI/ΔO	0.003 ± 0.00	0.003 ± 0.00	0.002 ± 0.00	0.002 ± 0.00	0.002 ± 0.00	0.003 ± 0.00	0.003 ± 0.00	0.003 ± 0.00
EI max/EI min (rEI)	5.39 ± 0.67	4.186 ± 0.31 *	4.96 ± 0.67	4.09 ± 0.41 *	5.61 ± 0.78	5.11 ± 0.69	4.39 ± 0.74 *	3.82 ± 0.14 *
O (EI max)/O min (rO)	1.9 ± 0.08	2.187 ± 0.14	1.95 ± 0.09	2.01 ± 0.06	1.82 ± 0.07	1.94 ± 0.09	2 ± 0.07	2.04 ± 0.03
rEI/rO	2.84 ± 0.41	1.93 ± 0.25 *	2.54 ± 0.32	2.03 ± 0.2 *	3.01 ± 0.45	2.64 ± 0.41	2.19 ± 0.34	1.87 ± 0.07 *

ΔEI: the difference between maximal and minimal EI values; ΔO: the difference between osmolality values at maximal and minimal EI; EI max/EI min: ratio of maximal and minimal EI values (rEI); O (EI max)/O min: ratio of osmolality values at maximal and minimal EI (rO). Means ± S.D.; * *p* < 0.05 vs. human.

## Data Availability

The data presented in this study are available on request from the corresponding author. The data are not publicly available due to ethical permission.

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
