# Peer review of "Interspecies Diversity of Osmotic Gradient Deformability of Red Blood Cells in Human and Seven Vertebrate Animal Species"

_cells, 2022, doi:10.3390/cells11081351_

Round 1

Reviewer 1 Report

Varga et al. presented the results of an extensive study of osmotic gradient deformability of RBCs in human and vertebrate animal species. Although the experimental material does not cause me any objections, it is necessary to draw the authors' attention to the following shortcomings in the presentation of the material.
1. It is necessary to provide in the "Introduction" a separate paragraph devoted to the motivation of this study.
2. The authors do not explain what caused this choice of animals.
3. It is not clear what the working hypothesis of the study is.
4. It is necessary to add a paragraph describing the "Limitations" of the presented study.

Author Response

Dear Reviewer,

thank you very much for the review with the opinion, which is an honor for us.

In the Introduction part we added the necessary sentences on the motivation, choice of animal species and working hypothesis:

"We hypothesized that osmotic gradient deformability of red blood cells may show a wider diversity in experimental/laboratory animal species that can be important in extrapolation of the results found in animal experiments versus human. The motivation of conducting a descriptive-comparative study was to reveal and better understand these micro-rheological differences, and their relation to other micro-rheological and hematological variables. When evaluating and compare the results, as an extrapolation, it is important to know the species-specific characteristics of these parameters. Therefore, our aim was to extend the investigations and support the osmoscan data with hematological and conventional RBC deformability measurements in human and seven mammalian animal species (pig, dog, cat, sheep, mouse, rat, and rabbit). These animal species are commonly or relatively commonly used in various experiments as small animal model or large animal model. To our knowledge, to date this is the most comprehensive comparative study on osmoscan parameters and directly related variables in a wide range of animal species compared to humans."

At the end of the Discussion part, a paragraph about the limitations has been included.

"Main limitations of this study includes the relatively low case numbers and certain methodological conditions. Since the shape of the osmoscan curves, and thus the values, are shear stress dependent [reference], it would be interesting to investigate these variables at various shear stress settings. Due to the limitation of the blood sampling volumes, we could not extend the investigations in this matter. In human studies, the set shear stress is generally 30 Pa (or 20 Pa), therefore for the comparison we decided to use one setting (30 Pa). Other limitation is the number of species. We selected mammalian species used in surgical research models. Obviously, a more wider investigation would give more valuable information about the characteristics and relation (even taxonomical) of the inter-species diversity. Further limitation is the potential effect of gender and age difference of the animal species that were investigated [references].”

Thank you very much again for the valuable review. We hope that the answers and corrections in the revised version might be acceptable.

Sincerely yours,

Norbert Nemeth

Reviewer 2 Report

A well written and executed study. Methodologies are well described and the results are well interrogated and interpreted. Minor point regarding presentation of the curves. It is difficult to differentiate between species/data sets. While descriptive in nature, the authors have presented a valuable data set and reference point for researchers in the biorheology field.

Minor grammatical point ln61-63 (correct small typographical errors).

(It is well known that osmolality affects RBC volume as cells are swelling 61 orshrinking, so affecting surface-to volume ratio and deformability, besides the importance of mechanical, elastic properties of the membrane). 

I would also like to see mention of prospective uses for these findings with respect to human clinical studies.

Author Response

Dear Reviewer,

thank you very much for the review with the opinion, which is an honor for us.

The typos have been corrected in the text. Concerning the prospective uses, in the Conclusion part we added these sentences:

"The findings might help the extrapolation and comparability of data with human. Furthermore, studying erythrocytes from various species having different red blood cell volume, shape and biomechanical features, may also provide information about abnormal cells’ micro-rheological features in human hematological disorders."

Thank you very much again for the valuable review. We hope that the answers and corrections in the revised version might be acceptable.

Sincerely yours,

Norbert Nemeth